# In situ synthesis of supported metal nanocatalysts through heterogeneous doping

No Woo Kwak [1], Seung Jin Jeong[1], Han Gil Seo[1], Siwon Lee[1], YeonJu Kim[1], Jun Kyu Kim[1], Pilgyu Byeon [1],
Sung-Yoon Chung [1] & WooChul Jung [1]

Supported metal nanoparticles hold great promise for many fields, including catalysis and renewable energy. Here we report a novel methodology for the in situ growth of architecturally tailored, regenerative metal nanocatalysts that is applicable to a wide range of materials. The main idea underlying this strategy is to selectively diffuse catalytically active metals along the grain boundaries of host oxides and then to reduce the diffused metallic species to form nanoclusters. As a case study, we choose ceria and zirconia, the most recognized oxide supports, and spontaneously form various metal particles on their surface with controlled size and distribution. Metal atoms move back and forth between the interior (as cations) and the exterior (as clusters) of the host oxide lattice as the reductive and oxidative atmospheres repeat, even at temperatures below 700 °C. Furthermore, they exhibit excellent sintering/coking resistance and reactivity toward chemical/electrochemical reactions, demonstrating potential to be used in various applications.

[1] Department of Materials Science and Engineering, Korea Advanced Institute of Science and Technology, 291 Daehak-ro, Yuseong-gu Daejeon 34141, Republic of Korea. These authors contributed equally: No Woo Kwak, Seung Jin Jeong. Correspondence and requests for materials should be addressed to W.J. (email: wcjung@kaist.ac.kr)

Metal nanoparticles (NPs) dispersed on the surface of functional oxides are widely used as heterogeneous catalysts for various reactions due to their high dispersion, large concentration of highly undercoordinated surface sites, and metal–support interaction that often greatly alters their catalysis[1–4]. However, metal NPs are unavoidably sintered at high temperatures (>500 °C), leading to severe losses in catalytic activity and selectivity[5–9]. Thus, the fabrication of supported metal particles with excellent high-temperature stability is an important challenge. Recently, the in situ growth phenomenon of metal NPs directly from an oxide support, known as 'ex-solution,' has been reported and has received much attention in the fields of high-temperature catalysis and renewable energy[10–22]. Taking the most studied perovskite oxides (ABO$_3$) as an example, when they serve as a hosting framework, transition or precious metals can be dissolved as cations in the B-site of the perovskite lattice under oxidizing conditions; they can also be ex-solved partially upon subsequent reduction as nano-sized metallic phases decorating the oxide surface. Compared to traditional particle synthesis and dispersion techniques, this process is faster, more cost-effective, and allows finer and better particle distribution. More importantly, its reversibility indicates that particle agglomeration can be avoided through re-oxidation, significantly enhancing the lifetime of the supported catalysts[10].

However, since the ex-solution is essentially a phase transition phenomenon on the surface of complex compounds, the choice of applicable materials including both metals and parent oxides to obtain metal NPs with the desired size distribution is very limited, whereas the catalytic properties are highly dependent on the material composition[23–25]. Moreover, most relevant studies have carried out reduction of the perovskite oxide at a highly elevated temperatures (mostly above 900 °C), mainly due to the difficulty associated with extruding the metal NPs. The need for such high annealing significantly limits the application of ex-solution particles[10–14,16–22].

Since a grain boundary contains more crystallographic defects compared to bulk lattice, it can accommodate a considerably larger amount of dopant species[26,27]. Thus, the range of material compositions (for both oxide support and supported metal particle) applicable to ex-solution can be widened by using the reducible transition metals that are present in excess in the grain boundary. Furthermore, the grain boundary serves as a fast highway for cation diffusion, e.g., the cation diffusion is much faster, often up to several orders of magnitude, than bulk diffusion[28–31], and as a favorable site for heterogeneous nucleation, allowing metal particles to be ex-solved readily at relatively low temperatures (≤700 °C).

Here we propose a new NPs synthesis strategy via heterogeneous doping of catalytically active metal cations along the grain boundaries of the host oxide. To verify the feasibility of this idea, we select pristine/doped CeO$_2$ and ZrO$_2$ as a host material and demonstrated that various metal NPs, including Ni, Co, Cu, Au and Pt, can be formed on their surfaces even at a reduction temperature of only 700 °C. Significantly, these particles still have the inherent advantages of ex-solution; they can be self-regenerated under oxidative and reductive conditions, while holding strong anchorage with the oxide surface to suppress the growth of metallic particles. Furthermore, we observed that NPs synthesized on ceria surface by this method exhibit excellent coking resistance and electrochemical reactivity toward CO chemical-oxidation and H$_2$ electro-chemical-oxidation.

## Results

**Synthesis of metal nanoparticles.** Figure 1 describes the overall synthesis processes to obtain metal nanoparticles (NPs) on an oxide support surface. It consists of two main steps. The first step is to feed metal sources into the grain boundaries of a host oxide. Here, we deposited metals on a sample surface by sputtering and then annealed the sample at 700 °C in air for the metals to diffuse into the host oxide. It is noteworthy that the metal species only penetrate the grain boundaries of the oxide support, and this will be referred to as "heterogeneous doping" in this manuscript. According to Harrison's classification[32], metal ions simultaneously diffuse into the bulk lattice and grain boundary at very high temperatures, but when the homologous temperature is reduced to about 1/3 of the melting point, only grain boundary diffusion occurs (known as Harrison's C-type kinetic regime). Accordingly, in the present study, the heat treatment was carried out at a temperature of 700 °C in air, given that the melting point of CeO$_2$ and ZrO$_2$ is 2475 °C and 2715 °C, respectively. It is noteworthy that these types of oxides are widely used[33–41] but difficult to apply the ex-solution because they possess exceedingly low solubility of transition metals[42,43]. We also evidently confirmed that the diffusion of Ni in the bulk grain interior is negligibly slow at 750 °C used in this experiment (Supplementary Figure 1). Indeed, with longer heat treatment, accordingly more metal sources pass through the oxide, and the time it takes to fully occupy the metal cations into the grain boundaries depends on the thickness of the sample. For example, it takes about 10 h to fully penetrate Ni into a 500 nm-thick, Sm-doped CeO$_2$ (Sm$_{0.2}$Ce$_{0.8}$O$_{1.9-\delta}$) thin film. Next, the second step is to partially reduce the heterogeneously doped sample to cause extrusion of metal NPs onto the surface. In this step, the metal layer remaining on the sample surface is etched away and the metal particles are then released by a reduction heat treatment. The choice of reducing atmosphere is very important. For example, an effective oxygen partial pressure ($pO_2$) that is low enough to reduce metal

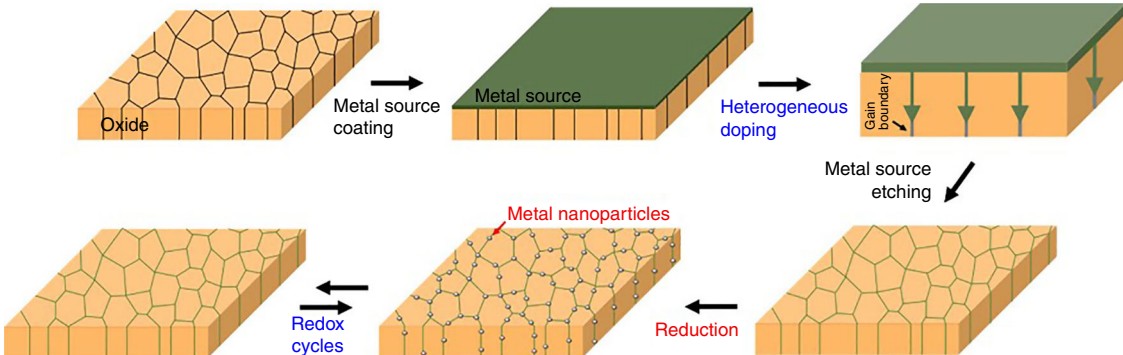

**Fig. 1** Synthesis of metal nanoparticles. Schematic diagram of the overall synthesis procedures of metal nanoparticles and their self-regenerative tendency according to redox cycles

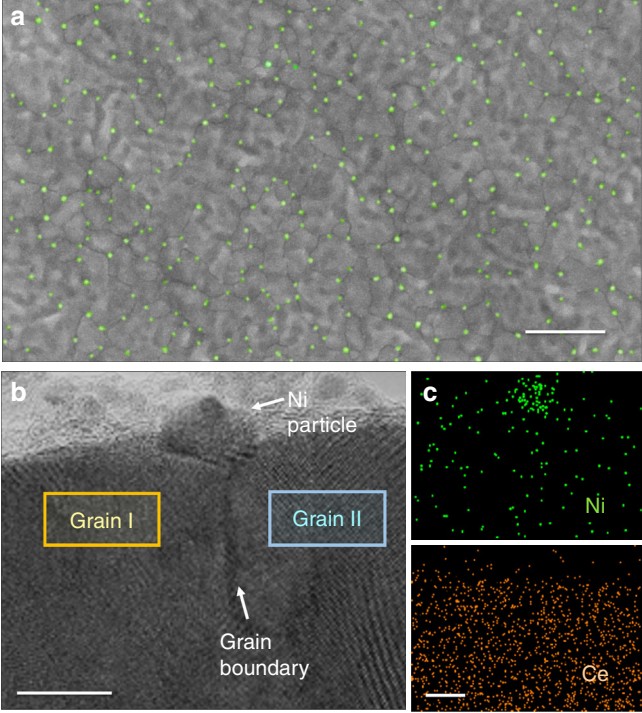

**Fig. 2** Microscopic images of metal nanoparticles on Sm-doped CeO$_2$ thin film. **a** Top-view field emission scanning electron microscopy (FE-SEM) image of synthesized Ni nanoparticles on Sm-doped CeO$_2$ (SDC) surfaces after annealing at 700 °C in reducing condition. For clarity, the nanoparticles were intentionally colored using Adobe Photoshop. **b** Cross-sectional scanning transmission electron microscope (STEM) image and (**c**) energy dispersive X-ray spectroscopy (EDS) mapping results of SDC film with a Ni particle. (Scale bars; (a) = 500 nm, (b) and (c) = 10 nm)

cations to neutral metals is required, while a sufficiently high $pO_2$ must be maintained so as to not decompose the host oxide.

**Physical and chemical characterization of nanoparticles**. Figure 2a shows field emission scanning electron microscopy (FE-SEM) images of a heterogeneously Ni-doped ceria thin-film sample after reducing at 700 °C and dry H$_2$ for 5 h. It becomes apparent that the monodisperse particles are uniformly distributed on the surface of ceria. The particle size distribution is very narrow, in this case 14 ± 3 nm. X-ray photoelectron spectroscopy (XPS) and energy-dispersive X-ray spectroscopy (EDS) results show that the particles are a Ni-rich phase, likely metallic Ni, which confirms that our proposed strategy works well (Supplementary Figure 2 and Supplementary Table 1). Ni NPs are mainly located at junctions where grain boundaries meet. This is because the grain boundary of the support is a repository and a diffusion pathway of Ni cations, and is also a site for Ni particle nucleation. Cross-sectional scanning transmission electron microscope (STEM) image and energy dispersive X-ray spectroscopy (EDS) maps of a grain-boundary region between two adjacent grains near the surface are shown in Fig. 2b, c. Fast Fourier transform (FFT) patterns also clarify that each of the grains has a mutually different crystallographic orientation, verifying the presence of a grain boundary between the two adjacent grains (see Supplementary Figure 3). It is clear that that the synthesized particle is Ni and it is deeply embedded in a grooved grain boundary junction between two different grains, with a wide particle-substrate interface. This is a structure that is commonly observed in ex-solution particles, indicating that the particles are strongly bound to the parent oxide[15].

We checked whether these particles were indeed generated by heterogeneous doping at grain boundaries. Cross-sectional scanning transmission electron microscope (STEM) images in Supplementary Figure 4 display that after heterogeneous doping, the diffused Ni species are selectively concentrated only at the grain boundaries of host ceria, whereas Ce atoms are spread over the entire areas. This observation implies that Ni NPs were formed from Ni species present inside the grain boundaries. This is further supported by secondary ion mass spectroscopy (SIMS) depth profiles; Ni preferentially enters across ceria grain boundaries after the oxygen heat treatment (Supplementary Figure 5b) and then escapes to the surface during the subsequent reduction process (Supplementary Figure 5c). The Ni particles observed here are not the species that remained on the surface due to insufficient chemical etching, which was confirmed by a XPS analysis of samples before and after etching (Supplementary Figure 2a).

Note that for the reliable analysis, a dense thin film of ceria have mainly been used as a support material in this study, but this method can also be applied to 3-dimensional porous structures (see Fig. 3a) and bulk polycrystalline pellets (see Supplementary Figure 6). Figure 3a shows the uniform distribution of Pt NPs on the surface of a nanostructured ceria thin film. In this case, Pt NPs were formed on the ceria surface by diffusion from a Pt source located below the ceria thin film (see Supplementary Figure 7).

**Comparison with ex-solution**. Unlike conventional ex-solution, our synthesis method has various advantages due to the inherent characteristics of grain boundaries. First, grain boundaries can contain much larger amounts of transition metal dopants relative to bulk interiors. This is generally accepted, although the migration mechanism of transition metals towards the grain boundaries of complex oxides has not yet been fully elucidated[44–51]. In this study, we measured the amount of transition metals (in this case Ni and Co) in the grain boundaries of CeO$_2$ thin films via Time-of-Fight Secondary Ion Mass Spectroscopy (ToF-SIMS) depth profiling and found that Co and Ni concentrations within the grain boundaries are almost two orders of magnitude greater than their bulk solubility (see Supplementary Figure 8). Therefore, the grain boundaries can serve as a reservoir for storing various transition metals, which makes it possible to ex-solve metals with very low solubility in a parent oxide bulk lattice by the heterogeneous doping strategy proposed in this study. As shown in Fig. 3b, we successfully formed Ni NPs on various pristine or doped CeO$_2$ and ZrO$_2$ supports. In addition to Ni, nanosized particles of various compositions such as Pt, Au, Co, and Cu could be synthesized (Fig. 3c). To the best of our knowledge, ex-solution of fluorite-structured oxides such as ceria and zirconia has rarely been reported[52–55] and, in particular, uniform dispersal of self-regenerative nanosized particles on their surfaces has not been achieved. Second, this method can greatly lower the reduction heat-treatment temperature required for ex-solution. Figures 2 and 3 show that metal NPs were readily formed at 700 °C, much lower than the range of 800–1000 °C typically reported for ex-solution of stoichiometric oxides with a fluorite structure. This is mainly because rapid cation diffusion through grain boundaries promotes the formation of NPs[56,57]. As shown in Fig. 4a, b, the longer the reduction heat treatment is, the more Ni precipitates emerge, and after a sufficient time, the size-distribution of particles no longer changes. For example, Supplementary Figure 9 shows how the size and number density of the Ni particles formed on the ceria surfaces change with the reduction time. At the beginning of the heat treatment (within approximately one hour), the particles grow and agglomerate at

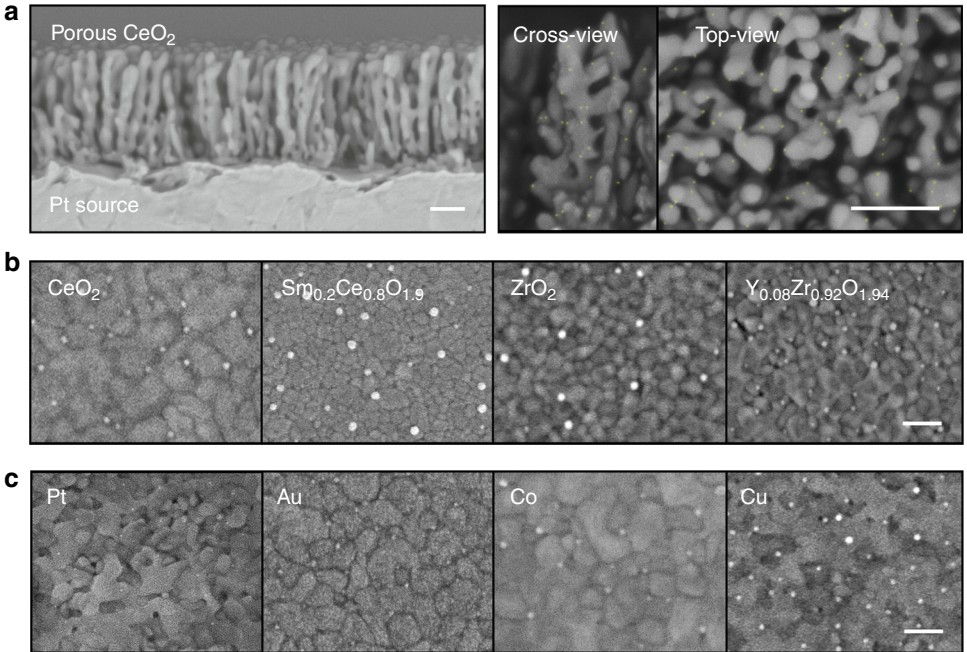

**Fig. 3** Microscopic images of samples with various compositions and structuresarious. **a** Low (left) and high-magnitude (right) field emission scanning electron microscopy (FE-SEM) images of synthesized Pt nanoparticles on a nanoporous $CeO_2$ thin film. For clarity, the nanoparticles are highlighted as yellow. Top-view FE-SEM images of synthesized (**b**) Ni nanoparticles on pristine/doped $CeO_2$ and $ZrO_2$ surfaces, and (**c**) Pt, Au, Co, and Cu nanoparticles on Sm-doped $CeO_2$ surfaces. (scale bars = 100 nm)

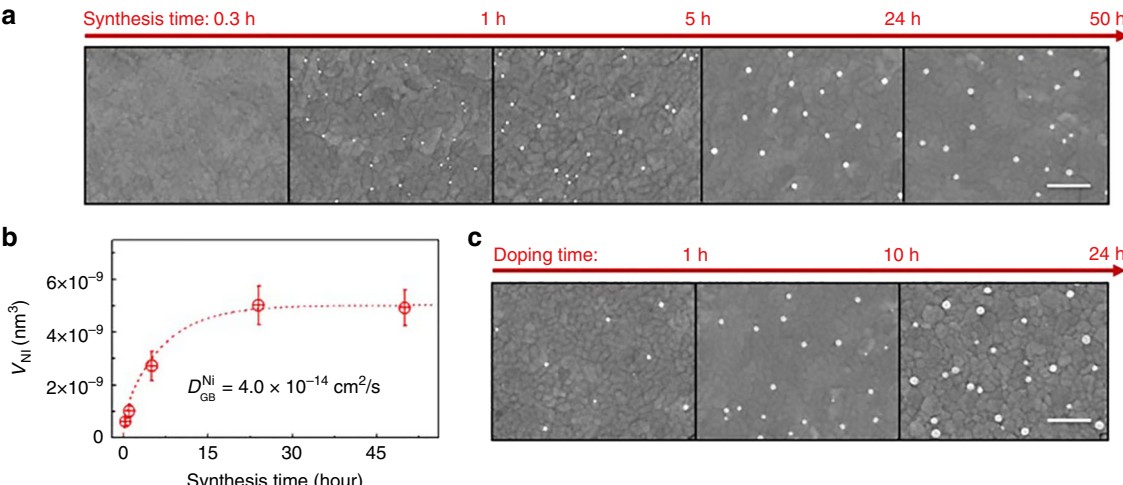

**Fig. 4** Evolution of ex-solved nanoparticles. The size-distribution (**a**) and total volume changes (**b**) of synthesized Ni nanoparticles on Sm-doped $CeO_2$ surfaces as functions of synthesis time and heterogeneous doping time (**c**) at 700 °C (scale bars = 300 nm)

the same time, but soon their size distributions become quite stable. This is a typical characteristic of heterogeneous nucleation. However, there is no change in the particle morphology after 24 h. This time evolution is a generally observed behavior limited by diffusion, where the amount of precipitated particles is approximately proportional to the square root of the reduction heat-treatment time (Fig. 4b). The corresponding diffusion coefficient of Ni in Sm-doped ceria was $4.0 \times 10^{-14}$ cm²/s at 700 °C, which is almost identical to the reported chemical diffusion coefficient of Ni through the grain boundaries of ceria[58]. The detailed derivation process of out-diffusion solution from finite source is described in Supplementary Equations 1–3. It should be noted that the obtained diffusion coefficient is ten orders of magnitude faster than that of a similar transition metal reported in ceria bulk

lattice (e.g., Fe in $Gd_{0.1}Ce_{0.9}O_{1.95}$[38]) when extrapolated to the same temperature (700 °C), and thus one can readily see the effectiveness of the grain boundaries for the migration of dopant cations. Lastly, the total amount of particles produced can be freely controlled by changing the supply of the metal source. As shown in Fig. 4c, with longer heterogeneous doping time, more metal is injected into the host oxide and thus a greater amount of metal NPs finally produced. It is also possible to control the grain size of the host oxide to change the particle size-distribution (see Supplementary Figure 10).

In addition to these unique properties, the particles synthesized in this study retain the inherent advantages of ex-solution: the ability to regenerate and to anchor strongly to the support. Here, we repeatedly annealed the synthesized Ni NPs in oxidizing/

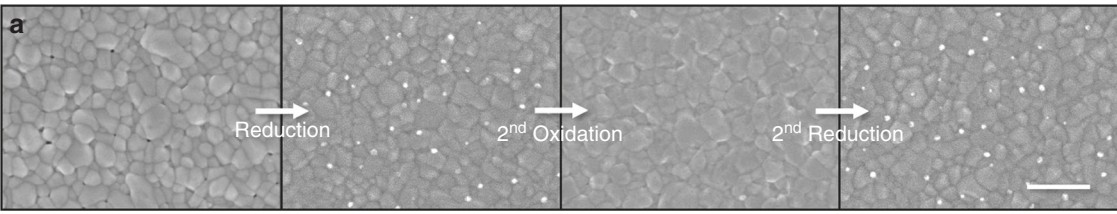

Synthesized by heterogeneous doping

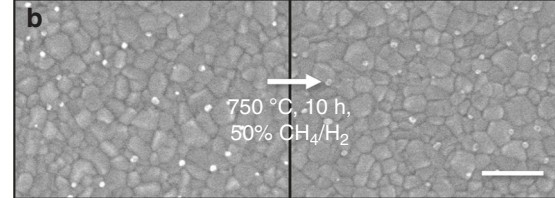

Synthesized by sputtering method

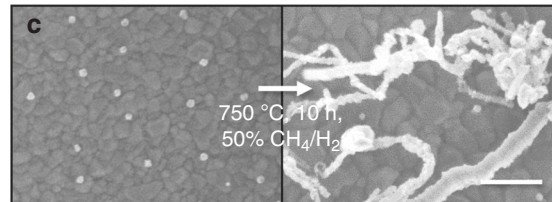

**Fig. 5** Regenerating and anti-coking ability of Ni nanoparticles. Field emission scanning electron microscopy (FE-SEM) images of Sm-doped $CeO_2$ (SDC) surfaces (**a**) according to redox cycles at 700 °C and after exposure to dry $CH_4$ with Ni nanoparticles synthesized by (**b**) heterogeneous doping and (**c**) sputtering method (scale bars = 300 nm)

reducing atmospheres and confirmed that self-regeneration of the particles was realized (Fig. 5a). In contrast, the Ni particles produced by sputtering for comparison continued to aggregate upon the same heat-treatment cycle. (Supplementary Figure 11). Furthermore, the strongly socketed Ni particles into the parent oxide show improved resistance to carbon deposition under a hydrocarbon atmosphere. Figure 5b, c indicate the difference in coking resistance between Ni NPs synthesized by heterogeneous doping and sputtering method at 750 °C and dry $CH_4$ condition.

**Catalytic reactivity of nanoparticles**. We tested the feasibility of the synthesized metal NPs on ceria as catalysts. Here, carbon monoxide oxidation $\left(CO + \frac{1}{2}O_2 \rightarrow CO_2\right)$ and hydrogen electro-oxidation $(H_2 + O^{2-} \rightarrow H_2O + 2e^-)$ were selected as examples of chemical and electrochemical reactions, respectively. First, we investigated the catalytic reactivity toward CO oxidation of Pt NPs on a nanoporous ceria surface (specific sample morphologies are in Fig. 3a). The conversion of CO via oxidation was tested with a fixed-bed flow reactor in the temperature range of 150–300 °C. The sample before each test was heat treated at 700 °C in the reactor, and the oxidizing and reducing atmospheres were repeated during the heat treatment to confirm the regenerating capability of the synthesized Pt NPs. The temperatures $(T_{10})$ at which 10% conversion of CO to $CO_2$ occurs according to the redox cycles are shown in Fig. 6a, and the right-off curves corresponding to each conversion result can be found in Supplementary Figure 12. It is apparent that the sample after the reductive heat treatment strongly activates CO oxidation, achieving nearly 100% conversion, whereas the oxidized sample is inert. These observations evidently confirm that Pt atoms move back and forth between the interior (as cations) and the exterior (as clusters) of the host ceria as the reductive and oxidative atmospheres repeat. Also, from the kinetic plots (Fig. 5b), we can see that the presence of Pt NPs on ceria surface improves the reaction rate of ceria by as much as 17 times at 210 °C with reduced $E_A$ by 0.3 eV.

Next, the electro-catalytic performance toward $H_2$ oxidation of the synthesized Ni NPs was investigated with AC impedance spectroscopy (ACIS) through the fabrication of symmetric cells composed of identical sized Sm-doped ceria thin-film electrodes on both sides of an yttria-stabilized zirconia single-crystal electrolyte. Nyquist plots of AC impedance results are shown in Fig. 6c (readers should refer to Supplementary Note 2 for specific ACIS analysis method). We observed that after reduction the

model electrode provided up to a 5-fold improvement in the electrode activity as evidenced by a decrease in the AC impedance resistance (Fig. 6d). For example, the electrode resistance values at 400 °C, $pH_2 = 1.5 \times 10^{-1}$ atm, and $pH_2O = 3.0 \times 10^{-3}$ atm were $4.0 \times 10 \,\Omega \,cm^2$ and $1.9 \times 10^2 \,\Omega \,cm^2$ for reduced and oxidized SDC, respectively. These observations clearly demonstrate the metal NP synthesis method proposed in this study could be useful in applications requiring catalysts with excellent durability and reactivity, such as high-temperature fuel cells, electrolyzers, sensors, catalytic converters, reformers, etc.

## Discussion
In summary, we successfully synthesized uniformly distributed metal NPs through selective doping of metal components into the oxide grain boundaries. The unique properties of the grain boundaries made it possible to solve the technical issues (i.e., limited range of applicable materials and high reductive heat treatment requirements) of ex-solution, an in situ growth phenomenon of metal NPs directly from a parent oxide. The metal particles synthesized in this study had the capability to regenerate upon redox cycles and high chemical durability, which are critical for prolonging the lifetime of the catalyst, as well as excellent chemical / electro-chemical catalytic activity. This approach is expected to present new possibilities for many applications in the field of catalysis and renewable energy.

## Methods
**Sample preparation**. $CeO_{2-\delta}$, $Sm_{0.2}Ce_{0.8}O_{1.9-\delta}$, $ZrO_{2-\delta}$, and $Y_{0.08}Zr_{0.92}O_{1.96-\delta}$ poly-crystalline thin films were prepared by means of pulsed laser deposition (PLD) from oxide targets of the respective materials on single crystalline $Al_2O_3$ (0001) substrates $10 \times 10 \times 0.65 \,mm^3$ in size (Dasom RMS, Korea). $CeO_{2-\delta}$ and $Sm_{0.2}Ce_{0.8}O_{1.9-\delta}$ targets were prepared by a combined EDTA–citrate complexing method[59] using $Ce(NO_3)_3 \cdot 6H_2O$ (JUNSEI Japan, 99.99% purity) and Sm $(NO_3)_3 \cdot 6H_2O$ (Alfa Aesar, 99.9% purity) precursors, and $ZrO_{2-\delta}$, and $Y_{0.08}Zr_{0.92}O_{1.96-\delta}$ targets were synthesized by a conventional solid state method using $ZrO_2$ (Sigma Aldrich, 99.99% purity) and $Y_2O_3$ (Sigma Aldrich, 99.99% purity) powders. A coherent COMPex Pro 205 KrF excimer laser, emitting at a wavelength of 248 nm, was used for ablation with the deposition parameters of a pulsed laser energy of 270 mJ and a laser repetition rate of 20 Hz. During the ablation process, the substrate temperature were kept at 500 °C, and the $O_2$ working pressures were set to 20 mTorr for dense films and 100 mTorr for porous films. After the deposition process, all of the samples were annealed in air in a tube furnace at 700 °C for 2 h to ensure that they were fully oxidized.

20 nm-thick $Co_3O_4$, NiO, and CuO layers and 200 nm-thick Pt and Au layers were deposited on ceria film surfaces by means of DC magnetron sputtering at a DC power of 100 W under controlled gas conditions ($Fe_3O_4$, NiO, CuO = 50% $O_2$/Ar, Pt, Au = 100% Ar). During the deposition step, the working temperature and pressure were maintained at room temperature and at 10 mTorr, respectively. All

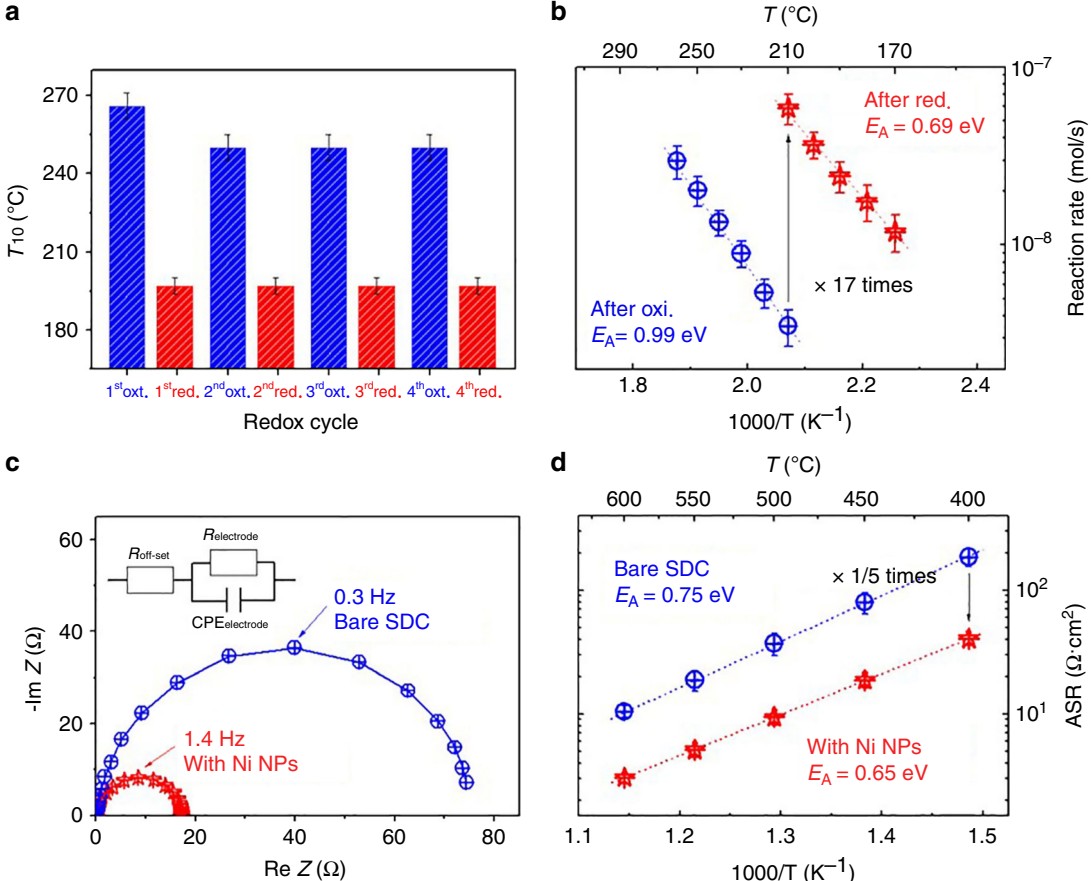

**Fig. 6** Chemical and electrochemical reactivity of Ni nanoparticles. **a** The temperatures at which 10% conversion of CO to $CO_2$ occurs on nanoporous $CeO_2$ thin films according to redox cycles at 700 °C. The combustion of CO via oxidation was measured at 250 °C, and the grain boundaries of ceria were heterogeneously doped with Pt. **b** Kinetic rate data for CO oxidation on the ceria thin films after oxidation/reduction at 700 °C. **c** Typical impedance spectra of SDC symmetric cells with fitting results denoted by the solid line at 60 °C (the ohmic resistance was removed for ease of the comparison of the electrode resistance). **d** A comparison of the temperature dependence of the electrode resistance for the $H_2$ electro-oxidation reaction after oxidation/reduction

diffusion sources were annealed after deposition in a tube furnace ($pO_2 = 0.21$ atm, Ar-balanced) at 300 °C for 5 h to crystallize them fully. After heterogeneous doping, the etching of the $Co_3O_4$, NiO, and CuO diffusion sources was undertaken in two steps. First, samples were reduced in a reducing atmosphere ($pH_2 = 0.05$ atm, Ar-balanced) at 600 °C for 2 h in a tube furnace, and the reduced surfaces of the Co, Ni, and Cu layers were subsequently etched by rinsing them in hydrochloric acid (Sigma Aldrich, 37%). A gold etchant (Sigma Aldrich, standard) was used for the etching of the Au diffusion source. After the etching step, the samples were rinsed in acetone, isopropanol, and deionized water in that sequence.

Patterned Ni|SDC|STF|YSZ|STF|SDC|Patterned Ni symmetric cells (SDC: $Sm_{0.2}Ce_{0.8}O_{1.9-\delta}$, STF: $SrTi_{0.95}Fe_{0.05}O_{2.975-\delta}$, YSZ: $Y_{0.16}Zr_{0.84}O_{1.92-\delta}$) for $H_2$ electro-oxidation experiment were also prepared by PLD on both sides of single crystalline YSZ (001) substrates $10 \times 10 \times 0.5$ mm$^3$ in size (MTI corporation). The laser operation was performed under the same conditions as the above dense films deposition experiment.

**Physical and chemical characterization**. Synthesized metal NPs were analyzed using a FE-SEM (SU8230, Hitachi) and XPS (K-alpah, Thermo VG Scientific) analyses. FE-SEM was operated at an acceleration voltage of 3 kV and an emission current of 2 μA without surface coating of samples, and back-scatter detector (BSE) was used in cross-section imaging for obtaining a clear image. The EDS scan in a SEM was conducted to obtain the chemical composition of particle with an accelerating voltage of 3 kV. The chemical characterization of particles was performed by XPS analysis with micro-focused monochromator X-ray source. The binding energy scale was calibrated by measuring the C1s peak at 285.0 eV, and the binding energy of Ni 2p was investigated. The depth profiles of Ni inside host oxides at each of experimental procedures were analyzed by SIMS (Caneca UNS 7f); primary 6 kV $Cs^+$ ions (9 kV of effective voltage) raster scanned over ($130 \times 130$ μm$^2$) were used to generate negatively charged secondary ions. A STEM (Talos F200x, FEI) analysis operated at 200 kV combined with EDS scan was carried out to identify the distribution of Ni inside host oxide. 4 windowless silicon drift detector (SDD) EDS system (Super X) was used as EDS detector. Phase-contrast

high-resolution electron microscopy (HREM) images were obtained using a transmission electron microscope (Libra200MC, Carl Zeiss) at 200 kV with a spherical aberration corrector (CEOS GmbH) for the objective lenses. The color of the nanoparticles in Fig. 2a was highlighted using Adobe Photoshop.

**Catalytic reactivity measurement**. A fixed-bed flow quartz reactor with a diameter of 1/2 inch was used to conduct the CO oxidation experiments over the temperature range of 150–300 °C. To build a catalytic bed, quartz wool was initially loaded into the middle of the reactor, and porous ceria (~500 nm thickness) | Pt (~250 nm thickness) | $Al_2O_3$ ($10 \times 10 \times 0.65$ mm$^3$ size) samples were loaded onto the quartz wool (a total of eight $Al_2O_3$ substrates were loaded for measurements). The temperature of the catalyst was measured using a K-type thermocouple in contact with the catalytic bed. The reactants consisted of 1% CO and 4% $O_2$ in Ar balance, and the total flow rate was adjusted to 50 mL/min through digital mass flow controllers (MFCs). The reactant and product gases were monitored using quadrupole mass spectroscopy (Pfeiffer Vacuum GSD320) in real time. The light-off curves were drawn with a ramping rate at 3 °C/min. The CO conversion ratio (%) was defined as $100 \times (mol_{CO, in} - mol_{CO, out})/mol_{CO, in}$. The signal of CO was calculated by considering the contribution of the cracking fragment of $CO_2$ (11.4%) in the mass concentration determination mode. After each measurement, the oxidation and reduction of the samples (700 °C for 10 h) were conducted in 21% $O_2$ and 5% $H_2$ in Ar balance, respectively.

Electrochemical analysis of Patterned Ni|SDC|STF|YSZ|STF|SDC|Patterned Ni symmetric cells were carried out by using ACIS (VSP-300, Biologic) under wet $H_2$ condition over the temperature range of 400–600 °C. AC amplitude of 10 mV was used throughout, after confirming that this voltage lies within the linear regime of the sample's current-voltage response. The measurement frequency range was from 0.01 Hz to 5 MHz. The cells were placed inside an alumina tube into which the mixtures of gas ($H_2$–$H_2O$–Ar/$H_2$–$O_2$–Ar) was delivered via digital MFCs. During measurement, the real-time humidity was monitored using a humidity sensor (Rotronic Hygroflex).

## Data availability

The data that support the findings of this study are available from the corresponding authors upon reasonable request.

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

## Acknowledgements

This study was supported by the Samsung Research Funding Center for Future Technology (G01150336).

## Author contributions

N.W.K. and W.J. conceived and designed the project. S.J.J., Y.K. and J.K.K. conducted the sample preparation. H.G.S. and S.L. measured the catalytic reactivity. P.B. and S.-Y.C. undertook TEM analysis. All authors contributed to discussions of the results.

## Additional information

**Competing interests:** The authors declare no competing interests.

