## [Peer Review File · Nature Communications]

Reviewer #1 (Remarks to the Author):

Please see attached document which contains my review.

Reviewer #2 (Remarks to the Author):

The paper describes a new concept whereby metal particles can be exsolved selectively and reversibly at the grain boundary of materials. The authors further demonstrate the utility of their concept for three different applications. I believe the concept of the paper is interesting and would be of interest to researchers in a variety of fields, beyond the growing field of exsolved materials. Overall, the text and figures are well presented and clear. The paper should be considered for publication in Nature Communications after addressing the issues outlined below.

Page 3, line 52. It is unclear what the authors mean by “A grain boundary with relatively open structure”, does this imply more crystallographic defects?

Page 4, lines 72-73. The authors need to provide more detail about how the metal layer was deposited to allow for others to reproduce the work.

Page 4, lines 74-75. The authors would need to provide more convincing evidence of this statement or provide alternative explanations. Currently this is only supported by the EDS images in Fig 2 which are not particularly clear or convincing and due to the nature of this technique only provide local details. Additionally, Fig S2c indicates that eg Ni dopes ceria throughout the bulk and not only in the grain boundary. Based on the data, I suspect that the Ni does substitute through the bulk structure to some extent, but only exsolves at the grain boundaries due to ease of nucleation and diffusion as compared to the bulk in the experimental conditions used in the paper.

Page 5, lines 86-87. The authors need to provide additional experimental details for the etching process.

Page 7, lines 115-116. In my opinion, this method would be difficult to apply to 3D porous structures due to the difficulty of coating such a sample. In fact, the author's own results in Fig. 3a illustrate this, it seems particles are present mostly on the top of the sample.

Page 7, lines 121-125. While it is correct that grain boundaries can store and release more metals than the bulk, at the same time it limits the sample area that can be decorated with particles, which means that this could limit the particle populations that can be achieved by this method. I believe the authors should at least attempt a comparison to set these results into a better context. For instance, from Fig 1 it is apparent that particle populations of <50 particles per square micron of sample area were achieved. A recent paper on exsolved materials illustrates that particle

populations between 150-700 particles per square micron can be achieved (Nature Communications, 8, 1855, 2017).

Page 8, lines 129-130. This is generally correct, although the authors seem to have missed a very relevant study, ACS Catalysis 2016, 6, 3688–3699.

Page 9, lines 133-134. This is incorrect, exsolution can be performed at temperatures as low as 550 C, see Nature Communications, 8, 1855, 2017.

Page 9, lines 136-137. In the respective figure there is also clear evidence of size evolution and possibly agglomeration. The authors should consider plotting the particle population as well as particle size versus time to help clarify this issue.

Page 11, line 172. The authors should plot the results for a temperature where conversion is not 100% because this data cannot be compared.

Page 11, line 173. Fig 7S needs to be replotted because in its current form cannot be read.

Page 11, lines 174-175. The details in which the catalytic studies were conducted are missing, eg amount of sample, flow rates, gas hour space velocities etc Currently it is impossible to put the results into context or to reproduce the experiments.

Reviewer #3 (Remarks to the Author):

In this paper the authors explore so called “exsolved” catalysts that are produced via the reductive egress of transition metal ion dopants that congregate at the grain boundaries a mixed ceria-based catalyst support. The concept is distinct from traditional exsolution catalysts where the catalytically metal dopants are soluble in the oxide host. I don't find the concept to be very practical since it is not clear how one could prepare a high surface area support where the catalytic metal atoms (ions) are initially present at the grain boundaries. In the work described here they have done this by first depositing a metal film on the surface of the oxide, diffusing the metal into the grain boundaries under oxidizing conditions at high temperature, and then removing the excess metal by etching the surface. The etching step is what makes this approach impractical.

While the synthesis method may not be generally applicable, it does produce a useful model system to study the role of grain boundaries in supported metal catalysts and they have obtained some

interesting results. The authors have made a convincing case that Ni ions can diffuse along the grain boundaries in Sm-doped CeO₂ under oxidizing conditions and then exsolve back to the surface upon reduction at high temperature. This latter step results in the formation of Ni nanoparticles on the surface of the oxide that are located at grain boundaries. These metal particles appear to be pinned to these locations. The reasons for this pinning is not clear. They argue that it is due to the particles being socketed into the surface as has been demonstrated for Ni particles formed by exsolution from the bulk of (La,Sr)TiO₃ in ref. 15. This may be occurring here, but the evidence for it is not very convincing and it is not clear how the particles would become embedded at the surface grain boundary. The only evidence they provide for the particles being embedded is the relatively low resolution SEM image in figure 2b which is difficult to interpret.

In my opinion the most interesting aspect of the work is that the exsolved Ni particles appear to be highly resistant to catalyzing the formation of carbon filaments when exposed to dry methane at 700C. This is similar to what has been reported in at least some studies of Ni particles that have been exsolved from the bulk of a host oxide. This anti-coking property would be advantageous in multiple chemical processes that use supported metal particles. .

Overall I found this to be an interesting fundamental study. Since it it's not clear how one could easily exploit the use of metal ion diffusion along grain boundaries as has been done here, the general applicability of the method may be limited. For this reason I'm not sure the paper meets all the criteria for publication in Nature Communications.

Responses to the Reviewers' Comments

We are pleased to submit the revised version of our manuscript, entitled “Heterogeneous doping, a novel in-situ synthesis of supported metal nanocatalysts” by N. W. Kwak, S. J. Jeong, H. G. Seo, S. Lee, Y. Kim, J. K. Kim, P. Byeon, S.-Y. Chung and W. Jung.

We thank the reviewers for their careful reading of the manuscript and appreciate their favorable view of the significance of our findings. We also thank the reviewers for their constructive comments and suggestions, which should help to improve the clarity of the manuscript.

Our responses to the reviewers' comments are given below. Also attached is the revised manuscript in which revisions in response to the reviewers' comments are indicated by use of the red text.

Reviewer #1

This a very important article. The work is novel in that it presents a strategy that can be applied to a new field (ex-solution) by systematically combining the "classical" concepts (i.e., heterogeneous nucleation or grain boundary diffusion). The authors also show large a amount of careful data that suggests that this strategy is applicable not only to 2D thin-film samples but also to 3D porous structures, indicating the potential for this strategy to be used in practical applications. Furthermore, the authors provide compelling scientific evidence that nanoparticle exsolution 1) is mediated by transition metal (TM) migration and storage at grain boundaries (GBs) and 2) takes places preferential phase separation at defective interfaces (e.g., GB-atmosphere interfaces). The authors provide a careful and thorough analysis of these issues. In light of its importance, scientific contribution, and potential applicability, I would like to suggest minor revisions for this relevant article. I have thought a few comments and questions, which I list below.

Authors:

We appreciate reviewer #1 for his or her very favorable evaluation of our research. We also thank the reviewer for acknowledging the outstanding novelty and potential applicability of our work. We do believe that the synthesis strategy proposed in this study is a useful and practical technique that greatly extends the applications of the emerging field known as ex-solution

Comments and questions:

1. It appears that the GBs act as a reservoir for the TMs. Could the authors explain/estimate what the maximum concentration/amount of TMs can be actually stored in the GB? What would be the maximum nanoparticle coverage?

Authors:

To answer this question by reviewer #1, we measured the amounts of transition metals (TMs, in this case Ni and Co) in the grain boundaries (GBs) of the CeO₂ thin films used in this study *via* time-of-flight secondary ion mass spectroscopy (ToF-SIMS) depth profiling. In order to determine the absolute values of the TM concentration directly from the intensities of the secondary ions, reference films containing Co or Ni with a known bulk concentration ($2.5 \times 10^{19}/\text{cm}^3$) were also used. Figure R1 exhibits the results of the intensity of the secondary ions corresponding to NiO⁻, CoO⁻, and CeO⁻ from heterogeneously doped ceria films and references. These results, together with the grain size and grain boundary width obtained through microscopic image analyses, enabled us to calculate the concentration of the TM saturated in ceria GBs (see equation R.1 below).

Fig. R1. Time-of-flight secondary ion mass spectroscopy (ToF-SIMS) depth profiles of secondary ions from (a) Ni-doped (0.1 at %) and (b) Co-doped (0.1 at %) reference samples, and heterogeneously (c) Ni-doped and (d) Co-doped ceria thin films

$$\begin{aligned}
 C_{TM}^{GB} &= C_{TM}^{bulk} \cdot \frac{I_{TM}^{hetero.}}{I_{TM}^{refer.}} \cdot \frac{V_{bulk}}{V_{GBs}} \\
 &= C_{TM}^{bulk} \cdot \frac{I_{TM}^{hetero.}}{I_{TM}^{refer.}} \cdot \frac{d}{2 \cdot \omega_{GB}} \quad , \quad (R.1)
 \end{aligned}$$

where C_{TM} is the concentration of the transition metal in the bulk or grain boundary, I_{TM} is the normalised secondary ion intensity of the respective species in the heterogeneously doped films or references, V is the estimated volume of the bulk or grain boundary, ω_{GB} is the grain boundary width, and d is the grain size.

Conclusively, the Co and Ni concentrations within the GBs are nearly two orders of magnitude greater than their bulk solubility values (Fig. R2). These observations support the contention that the grain boundaries can serve as a reservoir for storing various transition metals, which makes it possible to ex-solve metals even with very low solubility levels in a parent oxide bulk lattice using the heterogeneous doping strategy proposed in this study.

Fig. R2. Comparison of the solubility limits of transition metals (Co and Ni) in the bulk and grain boundaries of CeO₂ measured at 700 °C.

We also estimated what would be the maximum nanoparticle (NP) coverage on the ceria surface. Significantly, the surface coverage of metal NPs depends on the thickness of the film, the size of the grains and the grain boundaries, and the shape and diameter of the NPs. For example, suppose that Ni NPs with an average diameter of 20 nm and a hemispherical shape are synthesized on a ceria thin film with a thickness of 500 nm, a grain size of 100 nm, and a grain boundary width of 0.5 - 1 nm. In this case, if all of the Ni of the ceria film is extruded to the surface, particle coverage of approximately 20 to 40% can be expected. However, the image of the sample synthesized under similar conditions (Fig. R3) shows that the actual observed Ni NPs coverage is less than expected. This occurs because all Ni ions present in the grain boundaries during the reduction process are not

exsolved, as a large proportion remains in the film, as supported by the SIMS depth profile of the same sample (Fig. R3b).

Fig. R3. (a) Top-view field-emission scanning electron microscope (FE-SEM) image and (b) secondary ion mass spectroscopy (SIMS) depth profile of Sm-doped CeO₂ (SDC) surfaces after annealing at 700 °C for five hours in a reducing condition (scale bar = 500 nm)

2. It would be interesting to elucidate the mechanism of TM migration towards GBs. I suggest that the authors cite the literature beyond, refs 26 and 39, and perhaps provide guidance for future work in this crucial area.

Authors:

This is a good point. We agree that a mechanistic investigation of the segregation of TMs towards the GBs of oxides is a significant topic for future research. The selective accumulation of impurities or alloying elements at the grain boundaries of metallic alloys leads to the fracturing of the grain boundary as a result of temper brittleness, creep embrittlement, or grain boundary corrosion. Thus, GB segregation has long been studied in the field of metallurgy. Studies of the grain boundary segregation of metallic alloys have made many advances from earlier theories such as the Gibbs adsorption isotherm or the Langmuir-McLean isotherm to recent DFT-based models¹⁻⁷.

However, such models cannot directly be applied to oxide materials. One major difference between metals and metal oxides is in the charge states; these do not exist in metal alloys, which are uncharged mixed metals. For example, oxides contain negatively charged oxygen ions, which easily transform into the gas phase according to the working environments and change the oxidation state of the cations accordingly. In addition, in oxides, the formation of various types of defects (Schottky or Frenkel defects) and the transport of oxide ions or cations also often occur. In particular, the co-existence of cations and anions creates electrostatic interactions. These unique characteristics of oxide materials do not exist in metals. Therefore, it will be very challenging to clarify the GB

segregation reaction mechanism in oxides. We added a sentence containing the references mentioned above to the revised manuscript.

3. I am a bit puzzled by the sentence at line 115-116. The authors state that this method can be applied to 3D porous structures. Could the authors explain whether the mode of TM migration and nanoparticle egress is somewhat similar to that reported for the films? It appears to me that the TM migration and growth may be substantially different in this case.

Authors:

The synthesis process proposed in this study consists of two main steps: (1) feeding of the metal source by diffusion along the interface of the host oxide, and (2) the extrusion of metal NPs selectively at defective interfaces (e.g., GB-atmosphere interfaces) by partial reduction of the oxide. In this regard, we argue that the process of forming TM NPs in a 3D nanoporous structure is similar to that of dense thin films. As shown in Fig. R4, TM NPs are formed on the surface only when there are successive oxidation and reduction steps in both the porous and the dense samples, and no changes on the surfaces of any of the samples could be observed by the oxidation heat treatment alone. These observations indicate that similar migration and egress processes are occurring regardless of the morphology of the samples used in this work.

Of course, for oxide samples with a highly porous structure, TMs can readily migrate not only through the grain boundaries but also along the surfaces. However, in the 3D porous films used here, it is virtually impossible to separate the contributions of GB and surface migration. Nevertheless, because the surface and the GB are types of planar defects, it is clear that the basic nature of the synthesis process of the metal particles is identical.

Fig. R4. Surface field-emission scanning electron microscope (FE-SEM) images of porous CeO₂ samples (a) after an oxidation heat treatment alone and (b) successive oxidation and reduction treatments. The oxidation heat treatment was carried out at 700 °C in air for 20 hours, followed by a

reduction heat treatment at 700 °C dry H₂ for 10 hours (scale bars = 100 nm). For clarity, the nanoparticles were intentionally colored by the authors.

4. Could the authors explain why the ohmic resistance changes upon reduction (Fig. 6c)? I suggest that the authors subtract this value out in the final version of the article.

Authors:

We employ model symmetric cells which consist of patterned Ni thin films on YSZ single-crystal substrates. In our experience over the past few years, the impedance results of samples with nominally identical structures deviate by approximately 25% from each other⁸⁻¹⁴. The difference in the ohmic resistance shown in Fig. 6c is within 10%, which is a very convincing result considering that two different samples were compared. However, to clarify this for readers, as the reviewer noted, we modified Fig. 6c with regard to the impedance spectra by subtracting the offset resistance.

Fig. 6. (a) The temperatures at which 10% conversion of CO to CO₂ occurs on nanoporous CeO₂ thin films according to redox cycles at 700 °C. The combustion of CO *via* oxidation was measured at 250 °C, and the grain boundaries of ceria were heterogeneously doped with Pt. (b) Kinetic rate data for CO oxidation on ceria thin films after oxidation/reduction at 700 °C. (c) Typical impedance

spectra of SDC symmetric cells with fitting results shown by the solid line at 600 °C (the ohmic resistance was removed to simplify the comparison of the electrode resistance outcomes), and (d) the comparison of the temperature dependence of the electrode resistance for the H₂ electro-oxidation reaction after oxidation/reduction

Minor issues:

1. The sentence at lines 86-89 is not clear. I encourage the authors to revise it.

Authors:

The sentences at lines 86-89 were revised, as per the reviewer's suggestion.

“In this step, the metal layer remaining on the sample surface is etched away and the metal particles are then released by a reduction heat treatment. The choice of reducing atmosphere is very important. For example, an effective oxygen partial pressure (pO_2) that is low enough to reduce metal cations to neutral metals is required, while a sufficiently high pO_2 must be maintained so as to not decompose the host oxide.”

2. Size of nanoparticles. Could the authors explain how the scale was reported, and corresponding error bar (± 3 nm) obtained (line 95)?

Authors:

We used the image analysis software 'Image J' to measure the sizes of all the particles observed in Fig. 2a to obtain the average and standard deviation of the particles. The error bar (± 3 nm) indicates the standard deviation of the obtained particle size.

3. Line 110 is not clear, Ni enters CeO₂, through the entire material or preferentially across the GBs?

Authors:

The sentence at line 110 was revised, as per the reviewer's suggestion.

“Ni preferentially enters across the ceria grain boundaries after the oxygen heat treatment and then escapes to the surface during the subsequent reduction process.”

4. Fig. 5 caption – the last line should be “scale bar” not “scare bar.”

Authors:

The caption of Fig. 5 was corrected, as per the reviewer's suggestion. Thank you for your careful review.

5. Fig. S1-d should be a table. Also regarding the first column “CeM, SmM,” it is not clear to me why the authors have the M and L at the end of the element.

Authors:

The arrangement and caption of Fig. S1 (Fig. S2 in revised supplementary information) was modified, as per the reviewer’s suggestion.

Fig. S2. X-ray photoelectron spectroscopy (XPS) spectra from Sm-doped CeO₂ (SDC) surfaces after (a) etching of the sample surface, and (b) the extrusion of Ni nanoparticles, and (c) a field-emission scanning electron microscope (FE-SEM) image of a Ni nanoparticle on a ceria surface (scale bar = 50 nm).

Table S1 Energy dispersive spectroscopy (EDS) results of spots 1 and 2 from Fig. S2c.

	Spot 1 (at %)	Spot 2 (at %)
Ce	75	84
Sm	13	15
Ni	12	1

6. Fig. S2-c, could the authors explain why the concentration of Ni decreases at ~15 nm? It seems inconsistent with a purely diffusively driven process and with equation S3.

Authors:

The concentration of Ni near the ceria surface was increased sharply by the Ni NPs which formed on the surface, as opposed to it dropping to a depth of ~ 15 nm. This is supported by a significant reduction in the Ni intensity near the surface after only slightly etching the Ni particles, as shown in Fig. R5.

Fig. R5. Depth profiles of secondary ions after the reducing annealing step at 700 °C for three hours and the subsequent etching step in a hydrochloric acid (Sigma Aldrich, 37%) solution for 1 minute.

In addition, in the diffusion process expressed by equation S3, the concentration of the transition metal cation is used, which must be distinguished from the neutral metal concentration. As per the reviewer's suggestion, in the revised manuscript, we added a description to the caption of Fig. S2 (Fig. S5 in revised supplementary information) for clarify.

“**Fig. S5** Depth profiles of Ni^- , Al^- , SmO^- , and CeO^- from SIMS measurements (a) before and (b) after annealing in air, and (c) after the reduction process (The intensity of Ni^- increases near the top surface because Ni NPs exist on the film surface.)”

Reviewer #2 (Remarks to the Author):

The paper describes a new concept whereby metal particles can be exsolved selectively and reversibly at the grain boundary of materials. The authors further demonstrate the utility of their concept for three different applications. I believe the concept of the paper is interesting and would be of interest to researchers in a variety of fields, beyond the growing field of exsolved materials. Overall, the text and figures are well presented and clear. The paper should be considered for publication in *Nature Communications* after addressing the issues outlined below.

Authors:

We are delighted that the reviewer has considered our manuscript well organized and has recommended it to be published in *Nature Communications*. We also thank the reviewer for acknowledging the outstanding novelty and potential applicability of our work.

Page 3, line 52. It is unclear what the authors mean by “A grain boundary with relatively open structure”, does this imply more crystallographic defects?

Authors:

Yes, we did mean that a grain boundary is a planar defect. The corresponding sentence is now revised as per the reviewer’s suggestion.

“A grain boundary **with more crystallographic defects** compared to a bulk lattice can accommodate considerably more dopant species.”

Page 4, lines 72-73. The authors need to provide more detail about how the metal layer was deposited to allow for others to reproduce the work.

Authors:

The process of depositing the metal layer was accidentally omitted from the manuscript. We appreciate this point by the reviewer. Details pertaining to the metal layer deposition process are now included in the experimental section.

“20-nm-thick Co_3O_4 , NiO, and CuO layers and 200-nm-thick Pt and Au layers were deposited on ceria film surfaces by means of DC magnetron sputtering at a DC power of 100 W under controlled gas conditions (Fe_3O_4 , NiO, CuO = 50% O_2/Ar , Pt, Au = 100% Ar). During the deposition step, the working temperature and pressure were maintained at room temperature and at 10 mTorr, respectively. All diffusion sources were annealed after deposition in a tube furnace ($p\text{O}_2 = 0.21$ atm, Ar-balanced) at 300 °C for five hours to crystallize them fully.”

Page 4, lines 74-75. The authors would need to provide more convincing evidence of this statement or provide alternative explanations. Currently this is only supported by the EDS images in Fig 2 which are not particularly clear or convincing and due to the nature of this technique only provide local details. Additionally, Fig S2c indicates that eg Ni dopes ceria throughout the bulk and not only in the grain boundary. Based on the data, I suspect that the Ni does substitute through the bulk structure to some extent, but only exsolves at the grain boundaries due to ease of nucleation and diffusion as compared to the bulk in the experimental conditions used in the paper.

Authors:

In the heat treatment conditions used in this study, the diffusion of Ni through the ceria bulk lattice rarely occurred. In order to determine how much Ni was substituted through the ceria bulk lattice, we also measured the diffusion coefficients of Ni impurities into CeO₂ bulk pellets at a temperature range of 1250 to 1350 °C in air (see Fig. R6). The obtained depth profile of Ni impurities in the ceria bulk pellet after annealing at 1350 °C for 50 hours in air is shown in Fig. R6.

Fig. R6. (a) ToF-SIMS depth profiles of CeO⁻, SmO⁻, and NiO⁻ into a CeO₂ bulk pellet doped with 0.5 at % of Sm before and after a heat treatment at 1350 °C for 50 hours in ambient air, and (b) $I(\text{NiO}^-)/(\text{CeO}^-)$ and fitting result (solid line) as a function of the sputtering depth

The annealed sample shows an increased intensity level of NiO⁻ to a depth of 100 – 200 nm from the NiO/pellet interface. The determined intensity of NiO⁻ was normalized by that of CeO⁻ to ensure a constant reference point as a function of the depth from the NiO/pellet interface; this is replotted in Fig. R6 (b). The relative intensity profile of NiO⁻ shows a typical diffusion profile in polycrystals with two features: the first feature of bulk diffusion is ascribed to the sharp drop of over tens of nanometers near the surface, and the longer depth profiles at a deeper depth are attributed to the combination of rapid diffusion along the grain boundaries and slower bulk diffusion (Harrison-type B diffusion kinetics)¹⁵. Here, we determined the bulk diffusion coefficient using a modified solution

for diffusion from a slab source (the intensity of the surface NiO diffusion source was decreased after the heat treatment – with a finite source; see R.2 below)¹⁶.

$$I_{(x)} = a \left[\operatorname{erf} \left(\frac{x+h}{\sqrt{4D_b t + 4\sigma^2}} \right) - \operatorname{erf} \left(\frac{x-h}{\sqrt{4D_b t + 4\sigma^2}} \right) \right] + I_b, \quad (\text{R. 2})$$

Here, $I_{(x)}$ represents the normalized intensity of NiO⁻, a is a constant, x is the depth, h is the film thickness, D_b is the bulk impurity diffusion coefficient of the Ni ions, t denotes the annealing time, σ is the SIMS mixing parameter, and I_b is the normalized background intensity of NiO⁻. The h , σ , and I_b values were obtained by fitting the depth profiles of NiO⁻ from pre-annealed samples and were used to fit the results from annealed samples. The obtained diffusion coefficient of Ni inside the ceria bulk lattice in the temperature range of 1250 to 1350 °C is shown in Fig. R7.

Fig. R7 The values and activation energy of diffusion coefficient of Ni inside ceria bulk lattice in air.

Conclusively, we clearly confirmed that the diffusion of Ni in the bulk grain interior is negligibly slow at the temperature of 750 °C as used in this experiment. For example, the diffusion coefficient obtained between 1250 and 1350 degrees was extrapolated to a Ni migration length of only 0.1 nm for ten hours at 750 degrees. The diffusion experiments using bulk pellets were added to the modified supplementary information.

Page 5, lines 86-87. The authors need to provide additional experimental details for the etching process.

Authors:

Details about the etching of the metal layer have been added to the experimental section, as per the reviewer's suggestion.

“After heterogeneous doping, the etching of the Co_3O_4 , NiO, and CuO diffusion sources was undertaken in two steps. First, samples were reduced in a reducing atmosphere ($p\text{H}_2 = 0.05$ atm, Ar-balanced) at 600 °C for two hours in a tube furnace, and the reduced surfaces of the Co, Ni, and Cu layers were subsequently etched by rinsing them in hydrochloric acid (Sigma Aldrich, 37%). A gold etchant (Sigma Aldrich, standard) was used for the etching of the Au diffusion source. After the etching step, the samples were rinsed in acetone, isopropanol, and deionized water in that sequence.”

Page 7, lines 115-116. In my opinion, this method would be difficult to apply to 3D porous structures due to the difficulty of coating such a sample. In fact, the author's own results in Fig. 3a illustrate this, it seems particles are present mostly on the top of the sample.

Authors:

It appears that the reviewer misunderstood the process of fabricating the Pt-decorated 3D porous sample shown in Fig. R8. In this case, the Pt source was not coated on the top of the porous ceria film, but rather the oxide film was deposited on the Pt layer. In other words, a metal source is present below the porous oxide structure and thus Pt migrates from below to the surface of the oxide. Indeed, one of the main advantages of the proposed metal nanoparticle synthesis strategy is that it does not require a uniform metal source coating. It is possible to synthesize desired uniform metal NPs over the entire surface of the host oxide as long as the metal source is in contact with the oxide and it can diffuse heterogeneously through the oxide.

Moreover, as shown in both the cross-sectional and top-view images in Fig. R8, the synthesized Pt NPs are uniformly distributed both vertically and horizontally. It should be noted that the surface of the porous ceria shown in the image on the right is farthest away from the Pt source layer. For clarity, we highlighted the Pt NPs in yellow, as shown in Fig. R8 below.

Cross view

Top view

Fig. R8. The original and NP-highlighted scanning electron microscope (SEM) images are shown. NPs were highlighted using Photoshop 3.0 software (scale bars = 100 nm).

Page 7, lines 121-125. While it is correct that grain boundaries can store and release more metals than the bulk, at the same time it limits the sample area that can be decorated with particles, which means that this could limit the particle populations that can be achieved by this method. I believe the authors should at least attempt a comparison to set these results into a better context. For instance, from Fig 1 it is apparent that particle populations of < 50 particles per square micron of sample area were achieved. A recent paper on exsolved materials illustrates that particle populations between 150 - 700 particles per square micron can be achieved (Nature Communications, 8, 1855, 2017).

Authors:

Regarding this comment by the reviewer, we would like to emphasize the following two aspects. First, the main purpose of this study is to present the concept of synthesizing metal NPs *via* heteronomous doping, not to maximize the particle number density and the resulting catalytic performance. In keeping with our purpose, we intentionally used an SEM image (Fig. 1 in the original manuscript) of the sample containing Ni NPs, which are relatively large in size and small in number density, to distinguish the Ni particles clearly. Indeed, the number density of metal particles present on the host oxide surface can be controlled by reducing the grain size or increasing the heterogeneous doping time. Fig. R9 shows that decreasing the grain size from ~ 100 nm to ~ 10 nm greatly improves the number of Ni particles from approximately 50 particles/ μm^2 to nearly 250 particles/ μm^2 while holding the particle size similar. We also already demonstrated that the longer heterogeneous doping time, the higher number density of Ni NPs (see Fig. 4 (c)).

Fig. R9. Field-emission scanning electron microscope (FE-SEM) images of Ni nanoparticles on ceria surfaces at 700 °C synthesized for five hours as a function of the grain size of the host oxide (scale bar = 500 nm). Grain size: (a) = ~ 100 nm, (b) = ~ 10 nm

Fig. 4. (c) Scanning electron microscope (FE-SEM) images of synthesized Ni nanoparticles on ceria surfaces at 700 °C as a function of the heterogeneous doping time (scale bar = 300 nm)

Second, it is unfair to compare the ex-solution states between parent oxides with different compositions and structures. In Nature Communications 2017, as noted by the reviewer, highly A-site-deficient $\text{La}_{0.8}\text{Ce}_{0.1}\text{Ni}_{0.4}\text{Ti}_{0.6}\text{O}_3$ and $\text{La}_{0.7}\text{Ce}_{0.1}\text{Co}_{0.3}\text{Ni}_{0.1}\text{Ti}_{0.6}\text{O}_3$ were used as host materials to promote the extrusion of Ni NPs. In contrast, it is well known that to exsolve particles with the stoichiometric oxide with a fluorite structure as discussed here is more difficult. For example, references 48 and 50 in the revised manuscript contain only a small number of NPs ($< 5 \text{ \#}/\mu\text{m}^2$) on the ceria surfaces.

As the reviewers pointed out, the synthetic method proposed in this study is dependent on the grain boundaries of the host oxide, which may be a technical constraint. However, as mentioned above, it is possible to synthesize uniform particles only by limited contact of the metal source. Furthermore,

because GBs have a substantially higher transition metal storage capacity than the bulk, it is also possible to produce an adequate amount of metal particles with a limited grain boundary density. Thus, we believe that this concept can be applied to a wide range of applications when it is properly combined with various techniques for fabricating nanocrystalline oxides.

Page 8, lines 129-130. This is generally correct, although the authors seem to have missed a very relevant study, ACS Catalysis 2016, 6, 3688–3699.

Authors:

We are delighted about reviewer's cautious comment. We added the reference (ACS Catalysis 2016, 6, 3688–3699) in page 8, lines 129-130.

Page 9, lines 133-134. This is incorrect, exsolution can be performed at temperatures as low as 550 C, see Nature Communications, 8, 1855, 2017.

Authors:

We agree with the reviewer's point. As described above, the tendency of the ex-solution greatly changes according to the composition and structure of the material; thus, the temperature at which the ex-solution occurs also varies widely from material to material. We revised the sentence at the request of reviewer #2.

“Figs. 2 and 3 show that metal NPs were readily formed at 700 °C, much lower than the range of 800–1000 °C typically reported for the ex-solution of stoichiometric oxides with a fluorite structure.”

Page 9, lines 136-137. In the respective figure there is also clear evidence of size evolution and possibly agglomeration. The authors should consider plotting the particle population as well as particle size versus time to help clarify this issue.

Authors:

Fig. R10 (Fig. S9 in revised supplementary information) shows how the size and number density of the Ni particles formed on the ceria surfaces change with the reduction time. At the beginning of the heat treatment (within approximately one hour), the particles grow and agglomerate at the same time, but soon their size distributions become quite stable. This is a typical characteristic of heterogeneous nucleation. We added plots to clarify this to the revised manuscript as per the reviewer's suggestion.

Page 9, lines 136-137: As shown in Figs. 4a-b, for a longer reduction heat treatment, more Ni precipitates emerge, and after a sufficient time, the size distribution of the particles no longer changes.

For example, Fig. S9 shows how the size and number density of the Ni particles formed on the ceria surfaces change with the reduction time. At the beginning of the heat treatment (within approximately one hour), the particles grow and agglomerate at the same time, but soon their size distributions become quite stable. This is a typical characteristic of heterogeneous nucleation.

Fig. R10. The changes in the (a) size and (b) number density of Ni NPs as a function of the synthesis time at 700 °C

Page 11, line 172. The authors should plot the results for a temperature where conversion is not 100% because this data cannot be compared.

Authors:

Page 11, line 172 and Fig. 6 (a) were modified as per the reviewer's suggestion.

Page 11, line 172: The temperatures (T_{10}) at which 10% conversion of CO to CO₂ occurs according to the redox cycles are shown in Fig. 6a.

Fig. 6. (a) The temperatures at which 10% conversion of CO to CO₂ occurs on nanoporous CeO₂ thin films according to redox cycles at 700 °C. The combustion of CO *via* oxidation was measured at 250 °C, and the grain boundaries of ceria were heterogeneously doped with Pt. (b) Kinetic rate data for CO oxidation on the ceria thin films after oxidation/reduction at 700 °C. (c) Typical impedance spectra of Sm-doped CeO₂ (SDC) symmetric cells with fitting results denoted by the solid line at 600 °C (the ohmic resistance was removed for ease of the comparison of the electrode resistance), and (d) a comparison of the temperature dependence of the electrode resistance for the H₂ electro-oxidation reaction after oxidation/reduction.

Page 11, line 173. Fig 7S needs to be replotted because in its current form cannot be read.

Authors:

The figure was replotted, as per the reviewer. (Fig. S12 in revised supplementary information)

Fig. S12. Catalytic activity curves of CO₂ conversion with respect to the temperature for porous ceria according to the redox cycle

Page 11, lines 174-175. The details in which the catalytic studies were conducted are missing, eg amount of sample, flow rates, gas hour space velocities etc Currently it is impossible to put the results into context or to reproduce the experiments.

Authors:

The processes of the catalytic tests were accidentally omitted from the manuscript. We appreciate this point by the reviewer. Details about them are given below and are included in the revised manuscript.

Page 15, line 239: A fixed-bed flow quartz reactor with a diameter of 1/2 inch was used to conduct the CO oxidation experiments over the temperature range of 150 – 300 °C. To build a catalytic bed, quartz wool was initially loaded into the middle of the reactor, and porous ceria (~ 500 nm thickness) | Pt (~ 250 nm thickness) | Al₂O₃ (10 x 10 x 0.65 mm³ size) samples were loaded onto the quartz wool (a total of eight Al₂O₃ substrates were loaded for measurements). The temperature of the catalyst was measured using a K-type thermocouple in contact with the catalytic bed. The reactants consisted of 1% CO and 4% O₂ in Ar balance, and the total flow rate was adjusted to 50 mL/min through digital mass flow controllers (MFCs). The reactant and product gases were monitored using quadrupole mass spectroscopy (Pfeiffer Vacuum GSD320) in real time. The light-off curves were

drawn with a ramping rate at 3 °C / min. The CO conversion ratio (%) was defined as $100 \times (\text{mol}_{\text{CO, in}} - \text{mol}_{\text{CO, out}}) / \text{mol}_{\text{CO, in}}$. The signal of CO was calculated by considering the contribution of the cracking fragment of CO₂ (11.4%) in the mass concentration determination mode. After each measurement, the oxidation and reduction of the samples (700 °C for ten hours) were conducted in 21% O₂ and 5% H₂ in Ar balance, respectively.

Reviewer #3 (Remarks to the Author):

In this paper the authors explore so called “exsolved” catalysts that are produced via the reductive egress of transition metal ion dopants that congregate at the grain boundaries a mixed ceria-based catalyst support. The concept is distinct from traditional exsolution catalysts where the catalytically metal dopants are soluble in the oxide host.

Authors:

We are pleased that reviewer #3 acknowledges that the proposed idea is distinct from the concept of the conventionally reported ex-solution technique. This is a novel aspect of this study that we wanted to emphasize.

I don't find the concept to be very practical since it is not clear how one could prepare a high surface area support where the catalytic metal atoms (ions) are initially present at the grain boundaries. In the work described here they have done this by first depositing a metal film on the surface of the oxide, diffusing the metal into the grain boundaries under oxidizing conditions at high temperature, and then removing the excess metal by etching the surface. The etching step is what makes this approach impractical.

Authors:

With regard to the practicality and applicability of the proposed metal NP synthesis strategy as pointed out by the reviewer, we would like to emphasize the following two points. First, the diffusion of transition metals through the GBs of a complex oxide is much faster than through bulk lattices, and the GBs have the ability to contain TMs higher than one order of magnitude over the bulk. Thus, this extreme selectivity for heterogeneous doping allows the desired catalytic metal ions to be present at the GBs of the oxide supports. We believe that this coupled with a number of state-of-the-art synthesis techniques for fabricating porous nano-polycrystalline oxide structures with high specific surface areas makes the approach proposed in this work reasonably practical.

Second, unlike reviewer #3's claim that the etching step makes our approach impractical, this synthetic technique does not require the complete removal of the metal sources. As mentioned in the earlier answer to the comment from reviewer #2, even with limited contact with a metal source, it is possible to diffuse the metal ions uniformly into all GBs within the host oxide if only a concentration gradient is present. Thus, by properly controlling the coating coverage of the metal source, an oxide support can be created in which metal particles are evenly distributed on the surface without any etching step. The only reason we carefully etched the metal on the sample surface in this study was to eliminate the possibility of NPs being formed from the metal source remaining on the surface. Indeed, in practical applications, the particles additionally generated from the residual surface metal source are not a problem, as they can also act as a catalyst.

In summary, because the proposed technique is based on grain boundary diffusion with excellent selectivity, (1) it does not require uniform coating of a metal source, (2) targeted metal ions can be uniformly distributed in the grain boundaries of the oxide only with an appropriate heat treatment, and (3) it does not require complete etching of the metal source afterward. These three aspects demonstrate that our approach can be applied to a variety of fields in practice.

While the synthesis method may not be generally applicable, it does produce a useful model system to study the role of grain boundaries in supported metal catalysts and they have obtained some interesting results. The authors have made a convincing case that Ni ions can diffuse along the grain boundaries in Sm-doped CeO₂ under oxidizing conditions and then exsolve back to the surface upon reduction at high temperature. This latter step results in the formation of Ni nanoparticles on the surface of the oxide that are located at grain boundaries. These metal particles appear to be pinned to these locations. The reasons for this pinning is not clear. They argue that it is due to the particles being socketed into the surface as has been demonstrated for Ni particles formed by exsolution from the bulk of (La,Sr)TiO₃ in ref. 15. This may be occurring here, but the evidence for it is not very convincing and it is not clear how the particles would become embedded at the surface grain boundary. The only evidence they provide for the particles being embedded is the relatively low resolution SEM image in figure 2b which is difficult to interpret.

Authors:

First, as in the reviewer's comment, this is the very first scientific report to introduce how the concept of heterogeneous doping can be applied to supported metal catalysts; hence, we mainly focus on carrying out logical and convincing control experiments using a model system rather than on technical details. We are very pleased that the reviewer recognized this important point.

At per the request by the reviewer, an additional high-resolution TEM image analysis was conducted to provide clear evidence of the phenomenon by which the metal particles generated in this study were pinned at the grain boundaries. As shown in Fig. R12, the FFT patterns clearly show the boundary between two adjacent grains, and the EDS mapping confirms that Ni is strongly bonded to the GB-atmospheric interface.

a**b**
Fig. R12. Cross-section tunneling electron microscope (TEM) image and Energy Dispersive X-ray Spectroscopy (EDS) maps of adjacent grains near the surface: (a) The EDS map for each element along with the phase-contrast high-resolution episcopic microscopy (HREM) image verifies that Ni nanoparticles were pinned at the grain boundaries. (b) The enlarged HREM images (white rectangles) and corresponding fast Fourier transform (FFT) patterns of adjacent grains were clearly distinguished.

Indeed, these observations are the typical behavior of heterogeneous nucleation, which has long been studied. As already known from classical theories¹⁷⁻¹⁸, this phenomenon occurs because the energy barrier of the nucleation at the defect sites such as a GB is lower than at a free surfaces. For

example, as shown in Fig. R13, the energy barrier (or shape factor) is known to decrease as the number of defective interfaces increases in the order of the grain corners, grain edges, grain boundaries, and grain interiors (Fig. R13). Because the GB-atmosphere interface, which is the key nucleation site in the present study, has a much lower shape factor and thus an energy barrier as opposed to a free surface, it is thermodynamically explainable that the metal particles formed are pinned at this interface.

Fig. R13. The effects of wetting angle and nucleation site on the activation barrier for nucleation relative to homogenous nucleation

In my opinion the most interesting aspect of the work is that the exsolved Ni particles appear to be highly resistant to catalyzing the formation of carbon filaments when exposed to dry methane at 700C. This is similar to what has been reported in at least some studies of Ni particles that have been exsolved from the bulk of a host oxide. This anti-coking property would be advantageous in multiple chemical processes that use supported metal particles.

Authors:

Thank you for this good suggestion. We are also interested in the excellent anti-coking properties of the ex-solved Ni particles observed in this study, as this is a factor we hold to be very significant from a technical point of view.

Overall I found this to be an interesting fundamental study. Since it it's not clear how one could easily exploit the use of metal ion diffusion along grain boundaries as has been done here, the general applicability of the method may be limited. For this reason I'm not sure the paper meets all the criteria for publication in Nature Communications.

Authors:

As mentioned earlier, we believe that the concept proposed in this study is a practical technique that can be used in a variety of applications. Moreover, the focus of this study is not to optimize the technical details but to introduce the concept of heterogeneous doping for use with supported metal catalysts for the first time. Thus, the experimental results obtained by this process will be of great help to scholars in related fields. Nature Communications is a multidisciplinary journal covering a wide range of sciences, and results that can provide information about important advances of significance to experts in each field can be published without being restrictions related to academic or technical aspects. We therefore believe that this article satisfies the criteria of Nature Communications for publication.

References

1. McLean, D. Grain boundaries in metals. (Clarendon Press, New York; 1957).
2. Wynblatt, P. & Ku, R.C. Surface-energy and solute strain-energy effects in surface segregation. *Surface Science* **65**, 511-531 (1977).
3. Mukherjee, S. & Moranlopez, J.L. Surface Segregation in Transition-Metal Alloys and in Bimetallic Alloy Clusters. *Surface Science* **189**, 1135-1142 (1987).
4. Desu, S.B. & Payne, D.A. Interfacial Segregation in Perovskites.2. Experimental-Evidence. *Journal of the American Ceramic Society* **73**, 3398-3406 (1990).
5. Lovvik, O.M. Surface segregation in palladium based alloys from density-functional calculations. *Surface Science* **583**, 100-106 (2005).
6. Han, J.W., Kitchin, J.R. & Sholl, D.S. Step decoration of chiral metal surfaces. *Journal of Chemical Physics* **130** (2009).
7. Tran, R. et al. Computational study of metallic dopant segregation and embrittlement at molybdenum grain boundaries. *Acta Materialia* **117**, 91-99 (2016).
8. Chueh, W.C., Hao, Y., Jung, W. & Haile, S.M. High electrochemical activity of the oxide phase in model ceria-Pt and ceria-Ni composite anodes. *Nature Materials* **11**, 155-161 (2012).
9. Jung, W., Gu, K.L., Choi, Y. & Haile, S.M. Robust nanostructures with exceptionally high electrochemical reaction activity for high temperature fuel cell electrodes. *Energy & Environmental Science* **7**, 1685-1692 (2014).
10. Jung, W. & Tuller, H.L. Investigation of surface Sr segregation in model thin film solid oxide fuel cell perovskite electrodes. *Energy & Environmental Science* **5**, 5370-5378 (2012).
11. Seo, H.G., Choi, Y. & Jung, W. Exceptionally Enhanced Electrode Activity of (Pr,Ce)O_{2-δ}-Based Cathodes for Thin-Film Solid Oxide Fuel Cells. *Advanced Energy Materials* **8**, 1703647 (2018).
12. Jung, W. & Tuller, H.L. A New Model Describing Solid Oxide Fuel Cell Cathode Kinetics: Model Thin Film SrTi_{1-x}Fe_xO_{3-d} Mixed Conducting Oxides-A Case Study. *Advanced Energy Materials* **1**, 1184-1191 (2011).

13. Choi, Y., Brown, E.C., Haile, S.M. & Jung, W. Electrochemically modified, robust solid oxide fuel cell anode for direct-hydrocarbon utilization. *Nano Energy* **23**, 161-171 (2016).
14. Ji, S. et al. Integrated design of a Ni thin-film electrode on a porous alumina template for affordable and high-performance low-temperature solid oxide fuel cells. *Rsc Advances* **7**, 23600-23606 (2017).
15. Harrison, L.G. Influence of dislocations on diffusion kinetics in solids with particular reference to the alkali halides. *Transactions of the Faraday Society* **57**, 1191-1199 (1961).
16. Beschnitt, S. & De Souza, R.A. Impurity diffusion of Hf and Zr in Gd-doped CeO₂. *Solid State Ionics* **305**, 23-29 (2017).
17. Porter, D.A., Easterling, K.E., & Sherif, M.Y. Phase transformations in metal and alloys. (CRC Press, London; 1980).
18. Cahn, J.W. The kinetics of grain boundary nucleated reactions. *Acta Metallurgica* **4**, 449-459 (1956).

Reviewer #1 (Remarks to the Author):

The authors have greatly improved the manuscript and have addressed adequately my concerns.

Reviewer #2 (Remarks to the Author):

The authors addressed my comments. I recommend publication.

Reviewer #3 (Remarks to the Author):

In the revised version of the paper the authors have adequately addressed most of my concerns. I'm still not totally convinced that the synthesis method is practical, but the data they included in the response does argue against my point of view. I get their point that etching is not necessary to produce the particles pinned at the grain boundaries, but it is possible that the extra metal left on the surface will interact with the surface in a different way than that pinned at the GBs and may still produce coke when exposed to hydrocarbons, so this may still be a problem at least in some applications.

The higher resolution TEM data they provided in the response addresses my concern about whether the metal particles are socketed into the surface at the grain boundaries. I suggest that these data be included in the supplemental information file for the paper. Assuming that they do this, I am now comfortable recommending that the paper be accepted for publication.

Responses to the Reviewers' Comments

We thank the reviewers for their constructive comments and suggestions, which should help to improve the clarity of the manuscript. Our responses to the reviewers' comments are given below.

Reviewer #1

The authors have greatly improved the manuscript and have addressed adequately my concerns.

Reviewer #2

The authors addressed my comments. I recommend publication.

Authors:

We appreciate the favorable attitude of reviewers to our study. Your comments have helped us to further improve the manuscript.

Reviewer #3

In the revised version of the paper the authors have adequately addressed most of my concerns. I'm still not totally convinced that the synthesis method is practical, but the data they included in the response does argue against my point of view. I get their point that etching is not necessary to produce the particles pinned at the grain boundaries, but it is possible that the extra metal left on the surface will interact with the surface in a different way than that pinned at the GBs and may still produce coke when exposed to hydrocarbons, so this may still be a problem at least in some applications.

Authors:

I am glad that you understood the technical points that we mentioned. I also agree with your comments on the residual metal species that may adversely affect a certain reaction and hope to develop improved process technologies that will make the proposed concept more practical soon.

The higher resolution TEM data they provided in the response addresses my concern about whether the metal particles are socketed into the surface at the grain boundaries. I suggest that these data be included in the supplemental information file for the paper. Assuming that they do this, I am now comfortable recommending that the paper be accepted for publication.

Authors:

At the request of reviewer #3, we have now added all high-resolution TEM images to be included in the Supplementary Figure 3. Thank you for your constructive suggestion.